# Learning Multiple Data Distributions

## Abstract

Vulnerability to Distribution Shift is a major challenge for real-world applications of Deep Learning. Distribution Shift occurs when the test data distribution is not identical to the training distribution. We present Multi-Distribution Learning, exploring the effectiveness of Data Augmentation to prepare for Distribution Shift. We use common Data Augmentations to simulate multiple data distributions and test how many of these distributions can be learned with a single model. We first illustrate the impact of adding the test distribution to the training distribution, such that it is no longer a Distribution Shift. We find success with pairs of distributions, however, the model is limited when attempting to learn 10 distributions simultaneously. In order to search for a subset of distributions that train well together, we use a Lookahead search derived from research in Multi-Task Learning. From this analysis, we select 4 distributions with high affinities to train together and 1 negative distribution to use for performance comparison. These experiments confirm that the accuracy of models evaluated on a test set included in the training distribution is much higher compared to test distributions not included in training. We evaluate our distribution grouping with Zero-Shot Distribution Inference to a held-out positive distribution and the held-out negative distribution. We compare this Zero-Shot Distribution Inference to a strong baseline of the RandAugment training scheme, as well as to a weak baseline of a model trained without Data Augmentation. We conclude by discussing directions for future work in using Data Augmentation to combat Distribution Shift.

## 1 Introduction

Distribution Shift, defined as the performance on test distributions not identical to the training set, are the Achilles heel of Deep Learning. We propose a simple solution to use Data Augmentation for targeted Generalization, alleviating the problem of Distribution Shift. In training, we aim to cover as many data distributions as possible, such that a Distribution Shift is not completely novel to our model. We measure our models using the Average Distribution Accuracy and propose a new framework of Distribution Generalization Metrics. These performance scores are a weighted average of the model loss or accuracy, the number of distributions it can cover, and the diversity of these distributions.

We begin with two important questions for thinking about Distribution Shift. What are different data distributions? What does Distribution Shift in the real-world look like? A data distribution can be characterized based on the frequency of class labels. For example, the training set may contain 80% dogs and 20% cats, but the test distribution shifts to 40% dogs and 60% cats. In addition to distributions of labels, we can think about distributions of the inputs themselves. This is significantly harder to characterize due to the high-dimensional nature of raw inputs, such as pixel grids or word embeddings, or even intermediate vector representations from these sequential processing models.

The WILDS benchmark Koh et al. (2020) has been a very useful dataset to understand Distribution Shift in the real-world. Koh et al. collected examples of Distribution Shifts across applications in Wildlife Monitoring, Molecular Engineering, and Python Code Completion, to give a few examples. The authors characterize Distribution Shift into two categories, Subpopulation Shift and Domain Generalization. Subpopulation Shift is well illustrated by our previous example of a shifting frequency of cats and dogs. However, Subpopulation Shift can also refer to more high-level distributions, such as an increase in certain features of the input. Domain Generalization describes settings where the train and test domains are completely disjoint. For example, a sentiment classifier

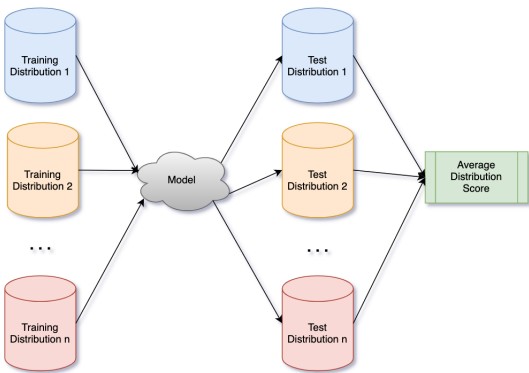

Figure 1: The Multi-Distribution Learning framework evaluates how training with a data distribution helps with test performance on that distribution. The framework tests the limits of how many distributions we can learn with a single model.

trained on movie reviews and then tested on restaurant reviews Gururangan et al. (2020), or an image classifier trained on photorealistic images and tested on paintings Peng et al. (2019).

Another technique for studying Distribution Shift is through corruption tests. Datasets such as ImageNet-C Hendrycks & Dietterich (2019), ImageNet-Corrupted, are constructed by applying common Data Augmentations to the original ImageNet test set. Models that achieve high performance on ImageNet typically do not generalize to ImageNet-C as well. There are several other examples of these datasets, such as smaller-scale CIFAR-C and MNIST-C tests Mu & Gilmer (2019), ImageNet-P with more subtle corruptions, and Stylized-ImageNet Geirhos et al. (2019) which uses a style transfer algorithm to disentangle content from style in images. Corruption testing, achieved by applying Data Augmentation to an original test set, is an important lens for viewing the phenomenon of Distribution Shift. We explore how we can similarly use Data Augmentation to prepare for Distribution Shift and the limits of this technique.

Our objective is to use Data Augmentation to target the Generalization we want the model to achieve. This is a simple, but promising strategy to inject prior knowledge about how the distribution will shift. We believe these assumptions on the test data distribution are necessary to make progress, inspired by the following quote from Arjovsky (2019), "If the test data is arbitrary or unrelated to the training data, then generalization is obviously futile." This quote supports the idea that we should make assumptions and bake in prior knowledge about the test distributions.

We begin our experiments by confirming that models which achieve high performance on an original test set will fail to generalize to the same test set corrupted with common data augmentations. This holds even for models trained with RandAugment Cubuk et al. (2020), a strong regularization strategy based on Data Augmentation. Our training strategy to achieve generalization to these test sets is illustrated in Figure 1. We apply N corruptions to the original data to form N training distributions. These N corruptions are additionally applied to the original test data to form an equivalent N test distributions. We report the average accuracy across the N test distributions. We begin by showing how this strategy increases the performance using N=2 augmentations.

We then scale our strategy up to learning N=10 distributions at once. We find that this requires much longer training times to achieve rivaling performance to single augmentation specializations trained with N=1. We borrow inspiration from research in Multi-Task Learning and use a Task Groupings algorithm Fifty et al. (2021) to search for distributions that train well together. We refer to these distributions as having a high affinity with each other. We use a Lookahead search to compare the performance of a proposed gradient update after training on a given distribution. Distributions are determined to train well together if the proposed update improves the performance across both test sets. We further attempt to disentangle the impact of repeating data in our distribution construction strategy by partitioning the CIFAR-10 set into 5,000 images each, to remove the confounding factor of repeating data. We find similar distribution groupings with and without repeating the original data.

We select 4 augmentations from our Lookahead search to train together. We additionally select 1 augmentation with a negative affinity for the sake of comparison. We find that the grouping achieves a much higher Average Distribution Score than the N=10 model trained with similar computation. We then generalize these experiments to a more realistic evaluation of Distribution Shift. We pose this as a problem of Zero-Shot Distribution Inference. We propose that groupings with a high affinity to the novel test distribution will perform better. We use a cross-validation analysis to iteratively pair 3 out of the 4 augmented distributions for training, evaluating the held-out distribution Zero-Shot. We find much better performance with this cluster compared to a strong RandAugment baseline, which is not completely Zero-Shot, as well as a weak baseline of a model trained without any Data Augmentation.

Our experiments utilize Data Augmentation to simulate diverse data distributions. We are currently limited to geometric transformations such as Rotations or Horizontal Flipping. However, we note that new frontiers in Data Augmentation such as CycleGAN Zhu et al. (2017) translations between domains or Text-to-Image generation Ramesh et al. (2021) should have a massive impact on this framework. We believe that Distribution Shift should be viewed similarly to Zero-Shot task inference commonly used in probing large models Liu et al. (2021). An interesting characteristic of Zero-Shot inference is the difference between massive models typically trained with less supervision than smaller, expert models. We view training on specialized distributions as analogous to supervised learning with particular tasks. Finally, we propose directions to extend our Distribution Generalization Score. Rather than the naive average accuracy, which we have reported in these experiments, we want a metric that rewards covering many distributions and the diversity of these distributions. We propose some ideas for measuring the diversity of these distributions.

In summary, we propose a framework of using Data Augmentation to anticipate and alleviate the challenges of Distribution Shift. Our contributions are as follows:

- We propose a new framework for Learning and Evaluating Multiple Data Distributions.
- We present a new metric for Distributional Generalization, the Average Distribution Score.
- We adapt a Task Grouping algorithm from Multi-Task Learning to study Multiple Distribution Learning.
- We present an analysis of Zero-Shot Distribution Inference, inspired by research on Zero-Shot Task Inference.

## 2 RELATED WORK

### 2.1 DISTRIBUTION SHIFT

Understanding the nature of Generalization is one of the core goals of Artificial Intelligence Chollet (2019). Many recent works have characterized types of Generalization. These include Domain Transfer Peng et al. (2019), Compositional and Systematic Generalization Johnson et al. (2016), Concept Drift, or Robustness, to give a few examples. Further characterizations include In-Distribution versus Out-of-Distribution Generalization, viewing Generalization gaps in a similar lens as fine- and coarse-grained classification. From a practical perspective, Koh et al. (2020) have published the WILDS benchmark characterizing the generalization needed for many real-world applications. In this work, we propose a simple framework for studying the relationship between training and test distributions. Andreassen et al. (2021) and Wortsman et al. (2021) have also explored the relationship between Zero-Shot robustness and fine-tuned models. Regardless of the particular type of generalization, we can likely form Data Augmentations that communicate expected Distribution Shifts.

### 2.2 CORRUPTION TESTING

Many works have found that Deep Learning systems are vulnerable to corruption tests. These tests are generally head-scratching cases because they very closely resemble the original input. This defies a natural intuition about the data and requires thinking closely about the distributions of inputs. Corruption tests range from injecting adversarial noise maps to applying Data Augmentations, as we have done in this work. Hendrycks & Dietterich (2019) presented ImageNet-C, a dataset constructed

from a set of 75 Data Augmentations and ImageNet-P constructed from adversarial injections. Ovadia et al. (2019) used corruption sets to evaluate Uncertainty techniques in Deep Learning. **?** utilized style randomization to understand intrinsic properties of Convolutional Neural Networks, finding a texture bias.

## 2.3 DATA AUGMENTATION

Multi-Distribution Learning is a technique building on Data Augmentation for improving the Generalization of Deep Neural Networks. Many works have covered different Data Augmentation techniques across many domains. Some examples include Kernel Filters, Geometric Transformations, Random Erasing, Color Space Transformations, Mixing Images, Adversarial Training, Neural Style Transfer, and GAN Data Augmentation Shorten & Khoshgoftaar (2019). In addition to individual augmentations, recent works have explored new strategies for utilizing Data Augmentation. This includes application in contrastive losses in self-supervised learning such as SimCLR Chen et al. (2020), as well as consistency losses in Unsupervised Data Augmentation Xie et al. (2020). Sinha et al. (2021) have additionally explored utilizing Data Augmentations that intentionally create Out-of-Distribution data. Our work is similar in that we are looking for new ways to utilize Data Augmentation in Deep Learning.

## 3 EXPERIMENTS

Our experiments report training and evaluation strategies for the ResNet152V2 He et al. (2016) on the CIFAR-10 dataset. The experiments are implemented using the Keras framework. The experiments are run on an NVIDIA A100 GPU in a Google Colab Pro+ runtime, with a maximum training time of roughly 20 hours. We experiment with up to 10 common Data Augmentations used for image data: Rotation, Translation, Crop, Cutout, Bright, Dark, Gamma Contrast, Horizontal Flip, Vertical Flip, and Jigsaw. These augmentations are implemented using the imgaug open-source library. Following is a quick description of how these augmentations transform the original data distribution.

The Rotation augmentation rotates an image along polar coordinates between 0 to 360 degrees. The Translation augmentation shifts an image along the x and y axes. This is done by moving pixels over/up by x pixels and adding 0s to displaced values. The Crop augmentation selects a subset of the original image and then resizes the subset to the original resolution with bilinear interpolation. The Cutout augmentation selects patches of the image and replaces the patch with 0 values. The Bright augmentation adds k to each pixel value in the image, essentially pushing it closer to 255 (white). The Dark augmentation inversely subtracts k to each pixel value in the image, essentially pushing it closer to 0 (black). The Gamma Contrast augmentation alters the distribution of pixel values to a smaller window between 0 and 255. The Horizontal Flip (LR Flip) augmentation mirrors the original image along the y-axis. The Vertical Flip (UD Flip) augmentation inversely mirrors the original image along the x-axis. The Jigsaw augmentation divides the image into n pixel windows and shuffles their location in the image.

As a quick summary of our experiments, Training with 2 Distributions characterizes the problem of Distribution Shift and the simple solution of adding the test distribution to the training set. The model is not training with the test set itself, just the underlying distribution that will be encountered in-the-wild. We simulate the different distributions through Data Augmentations. We then scale this up to N=10 distributions in training. Lookahead Search to Group Training Distributions utilizes Lookahead analysis to find distributions that complement each other during training. Results of Grouping Training Distributions compares the Average Distribution Score with the grouped distributions to N=1 and N=10 distribution training strategies. Table 4 illustrates the Zero-Shot Distribution Inference of the grouping with a strong RandAugment baseline and a weak baseline of training with No Data Augmentation.

## 3.1 TRAINING WITH 2 DISTRIBUTIONS

We begin by highlighting the problem of Distribution Shift, reproducing it in our experimental setup, and conducting simple experiments within the framework described in Figure 1. As shown in Table 1, when trained without any Data Augmentation the model performs much worse on the Rotation

| Training Technique | Original Test | Rotation Test | Jigsaw Test | Average Distribution Score |
|:---:|:---:|:---:|:---:|:---:|
| No Augmentation | 73.9% | 41.6% | 61.3% | 51.5% |
| RandAugment | **83.8%** | 55.0% | 69.6% | 62.3% |
| Rotation | 75.9% | **75.2%** | 63.2% | 69.2% |
| Jigsaw | 75.9% | 42.7% | 71.6% | 57.2% |
| Seq-RJ | 78.2% | 66.5% | 73.0% | 69.8% |
| Seq-JR | 75.9% | 71.5% | 69.4% | 70.5% |
| Alt-RJ | 80.0% | 73.7% | 74.8% | **74.3%** |
| Alt-SH-RJ | 80.2% | 73.3% | **75.0%** | 74.2% |

Table 1: The efficacy of training with a specialized distribution. The bolded values indicate the maximum accuracy achieved on each data distribution. The bolded values indicate the maximumm accuracy for each data distribution.

| Epoch | 10 | 30 | 50 | 200 | 350 | 1000 |
|:---:|:---:|:---:|:---:|:---:|:---:|:---:|
| Original Test | 65.5% | 68.4% | 72.5% | 77.0% | 79.8% | **83.4%** |
| Average Aug Score | 57.4% | 58.7% | 61.8% | 68.9% | 71.6% | **75.3%** |
| Max Aug Score | 78.5% | 77.1% | 78.0% | 79.2% | 79.5% | **83.2%** |
| Min Aug Score | 50.2% | 49.3% | 49.4% | 62.7% | 65.1% | **70.6%** |
| Max Aug Name | UD Flip | UD Flip | UD Flip | UD Flip | LR Flip | LR Flip |
| Min Aug Name | Rotate | Translate | Crop | Crop | Crop | Crop |

Table 2: Learning curves for Multi-Distribution Learning of 10 Augmentations. The schematic of this training strategy is shown in Figure 1.

and Jigsaw distributions compared to the original CIFAR-10 data. Even though the RandAugment training scheme dramatically improves performance on the original set, it is still susceptible to these augmentation derived distribution shifts. We then report the efficacy of specialized training on the distribution used in testing, such that it is no longer a Distribution Shift. We shift our aim to learning to specialize on multiple distributions in training. We consider two high-level schema for doing this. The first of which is Sequential Training (abbreviated as Seq in Table 1). Sequential Training refers to first learning the Rotation distribution and then learning the Jigsaw distribution. We encounter an interesting case of Catastrophic Forgetting where performance on the previously learned distribution quickly drops as it shifts focus to another data distribution. We find a better result with Alternating Training (abbreviated as Alt in Table 1). In Alternating Training we switch between each training distribution every epoch, rather than after convergence. We additionally test sharing a base feature extractor between each distribution and using separate classification heads to model each distribution (abbreviated as Alt-SH in Table 1). We do not find much difference with separate heads compared to fine-tuning the entire model. We continue with Alternating Training throughout the rest of the experiments.

## 3.2 TRAINING WITH 10 DISTRIBUTIONS

Following the success of learning 2 distributions in training, we set our sights on increasing the number of learned data distributions. Table 2 illustrates the learning curve of 10 data distributions. We find that it takes much longer for this strategy to achieve a strong Average Augmentation Score. However, after 1,000 epochs (roughly 20 hours of training with our computing setup described previously), the model achieves a fairly strong Average Augmentation Score with small variance and no significant minimum or maximum outliers in the accuracy scores. Throughout training the Rotation, Translation, and Crop Augmentations have the lowest performance, whereas Vertical and Horizontal Flipping (shown as UD Flip and LR Flip) have the highest score. At the end of training the multi-distribution learning model achieves a nearly identical score on the Horizontal Flip as the Original Test set, it could be interesting future work to see if the model has achieved invariance to this distribution shift in the prediction and representation spaces. We find more interesting details about the Vertical Flip augmentation in our analysis of Lookahead Search.

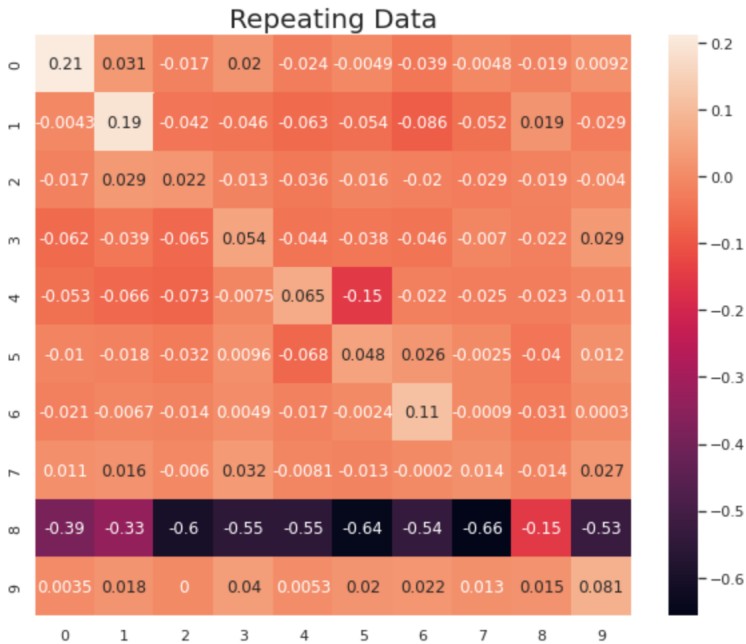

Figure 2: This table shows the performance difference by taking the proposed gradient update on the y-axis on the test distributions on the x-axis. In this index: Rotate = 0, Translate = 1, Crop = 2, Cutout = 3, Bright = 4, Dark = 5, Gamma Contrast = 6, LR Flip = 7, UD Flip = 8, and Jigsaw = 9.

## 3.3 LOOKAHEAD SEARCH TO GROUP TRAINING DISTRIBUTIONS

Due to the long training time of learning 10 distributions at once, we aim to find a subset of distributions that have a faster learning curve and higher performance. We adapt the Task Groupings algorithm Fifty et al. (2021) to search for this subset. This algorithm uses a Lookahead search to compute what the change in performance on other tasks, or distributions in our example, will change. For example, in Multi-Task Learning the Task Groupings algorithm computes the change in performance on Semantic Segmentation after taking a gradient update on Object Detection. We similarly compute the change in performance on the Translation distribution after taking a gradient step on the Rotated training set.

The results of the Distribution Grouping analysis are shown in Figure 2. The x-axis is the evaluated distribution and the y-axis is the distribution that proposes a gradient update. The augmentations are indexed according to their position in the following list: Rotate = 0, Translate = 1, Crop = 2, Cutout = 3, Bright = 4, Dark = 5, Gamma Contrast = 6, LR Flip = 7, UD Flip = 8, and Jigsaw = 9. We find a strong outlier result where a proposed gradient update on Vertical Flipping (UD Flip) causes a significant drop in performance across the other augmentations. This likely explains the optimization difficulty behind learning all 10 distributions simultaneously. The second strongest negative result is the impact of updating on Bright and evaluating on the Dark test distribution. We leave it to future work to explore if the inverse augmentations of increased and decreased Brightness have opposite gradient directions.

From Figure 2, we propose to group Rotation, Translation, Dark, and Jigsaw to form our subset. In order to further understand the impact of a gradient update on one distribution to the performance of another, we avoid repeating data. For example, the rotation distribution still has roughly the same image as the translation distribution in Figure 2. The right hand side of Figure 3 illustrates the lookahead updates when we split the data such that no distribution has the same original CIFAR-10 image. We find a similar grouping analysis with and without repeating data.

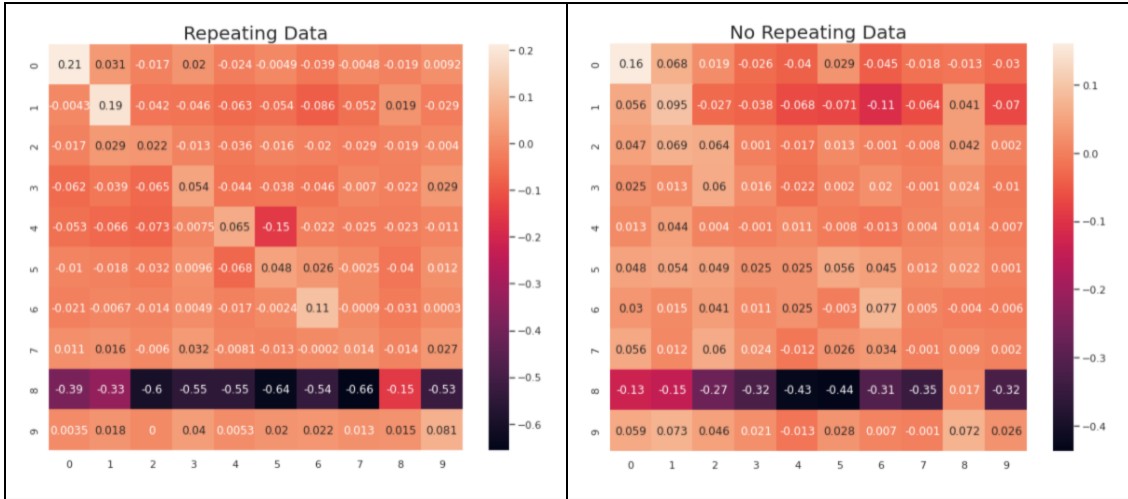

Figure 3: A Side-by-Side view of the impact of repeating data for Lookahead update analysis. Please see the caption of Figure 2 for more details on the x and y axes and the distribution indexing from 0 to 9.

## 3.4 RESULTS OF GROUPING TRAINING DISTRIBUTIONS

We analyze the performance of our distribution grouping on learning the training distributions and zero-shot inference to novel distributions. Table 3 reports enumerating through 3 of the 4 selected distributions for training. We encountered an optimization difficulty with the Rotate, Translate, Jigsaw (abbreviated as RTJ in Table3) that achieves low performance. However, the other 3 clusters outperform the 10 Augmentation scheme other than the Jigsaw test. We find an interesting division between distributions that benefit from specialization (Rotation and Translation) and others that benefit from the ensemble (Darker and Jigsaw). This could be an interesting property for the design of ensembles and specialist distribution models that we leave for future work.

The ultimate goal of learning multiple data distributions is to lessen the decrease in accuracy when faced with Distribution Shift. We test how the grouped distributions help with Zero-Shot Distribution Inference through a cross-validation analysis within the group. For example, we hold-out the Rotation distribution and train with Translate, Darker, and Jigsaw. The entires for 3 Augmentations in Table 4 reflect the Zero-Shot accuracy without training on that augmented distribution. For Rotation and Translation, we find a very strong Zero-Shot generalization with the group. As mentioned in Table 3, we ran into optimization difficulties with our run leaving out the Darker distribution and it achieves a very low accuracy. We found a similar performance on Zero-Shot Jigsaw generalization. We held-out Gamma Contrast as an example of a distribution with a negative affinity to the group. We report the best Zero-Shot accuracy on Gamma Contrast, supporting our analysis that the negative augmentation does not benefit from this particular cluster of training distributions. We compare this with the RandAugment model reported in Table 1, which achieves 83.8% accuracy on the original CIFAR-10 test set. Although RandAugment has never seen any of these distributions in isolation, it has seen compositions of them such as rotation, then translation, then darker applied to a single image. Thus, the RandAugment baseline is not truly Zero-Shot. The No Augmentation baseline highlights the poor performance when faced with Distribution Shift. In both RandAugment and No Augmentation we see the lowest performance decrease with the weakest augmentation, Darker. This highlights the need to develop distribution distance scores to weight the performances.

## 4 DISCUSSION

### 4.1 FUTURE DIRECTIONS FOR SIMULATING DATA DISTRIBUTIONS

Our experiments tested simple data augmentations to simulate distribution shifts. We have used the terms "augmentations" and "distributions" interchangeably to propose their similarities. We

| Training Distributions | Rotate Test | Translate Test | Darker Test | Jigsaw Test | Average Distribution Score |
|---|---|---|---|---|---|
| 3 Augs (TDJ) | N/A | 72.4% | **82.2%** | 67.4% | 74.0% |
| 3 Augs (RDJ) | 72.7% | N/A | 81.0% | 66.3% | 73.3% |
| 3 Augs (RTJ) | 37.3% | 33.2% | N/A | 42.2% | 37.6% |
| 3 Augs (RTD) | 71.9% | 71.0% | 81.4% | N/A | 74.8% |
| 10 Augs, 100 Epochs | 54.3% | 49.3% | 62.7% | 57.6% | 56.0% |
| 10 Augs, 1000 Epochs | 70.9% | 74.8% | 76.3% | **76.0%** | **74.5%** |
| 1 Aug | **75.2%** | **79.4%** | 74.9% | 71.6% | N/A |

Table 3: Comparison of Generalization to Test Distributions included in the Training Distribution.

| Training Distributions | Rotate Test | Translate Test | Darker Test | Jigsaw Test | Gamma Contrast Test |
|---|---|---|---|---|---|
| 3 Augmentations | **72.4%** | **71.9%** | 45.5% | 65.2% | 62.6% |
| RandAugment | 57.4% | 49.7% | **82.2%** | **69.5%** | **72.6%** |
| No Augmentation | 46.5% | 35.6% | 68.1% | 62.5% | 54.2% |

Table 4: Zero-Shot Distribution Inference.

believe techniques such as CycleGAN Zhu et al. (2017) and DALL-E Ramesh et al. (2021) will allow us to control Domain Generalization with Data Augmentation-style interfaces. For example, the iWildlife-monitoring camera in the WILDS benchmark encounters a new location for a wildlife monitoring camera at test time. A CycleGAN may be able to map the existing training data into this new location, further guided with text interfaces from DALL-E. We propose that we can train with this inferred novel distribution to perform better in the Distribution Shift.

## 4.2 ZERO-SHOT DISTRIBUTION INFERENCE

Few-Shot Learning describes learning a new task with a few examples, hopefully achieved by leveraging information from previously learned tasks. Zero-Shot Learning refers to a more extreme case in which the model performs a new task, but no additional learning is allowed. We can similarly view Distribution Shifts as having a few examples from the new distribution or having to perform Zero-Shot task inference. Another interesting trend in Few- and Zero-Shot learning is the efficacy of larger-scale models. Models such as GPT-3 can be guided with prompts to perform novel language tasks. It remains to be seen whether collections of expert models assigned to their respective inferences will outperform a single monolithic model. Even if it becomes unnecessary to train expert models for collections of tasks, or distributions as we have argued, understanding novel tasks will still be useful for evaluation.

## 4.3 DISTRIBUTION GENERALIZATION SCORE

Our experiments report a simple average augmentation accuracy. For future work we intend to explore metrics that better capture the diversity of a model's generalization ability. We intend to weight the average accuracies by the number of evaluated distributions and the distance between these distributions. The distance between distributions is a challenging quantity to measure. One solution, similar to ideas in the automated evaluation of generated images such as the Inception Score, could be to embed each distribution into the feature space of a pre-trained classifier and use these vector representations to compute distances. A strong Distribution Generalization Score should also be correlated with performance on Zero-Shot Distribution Inference.

## 5 CONCLUSION

In Conclusion, we have presented Multi-Distribution Learning, a framework to study the impact of training on potential test distributions. This is a simple framework to leverage the prior knowledge about how a data distribution is likely to change in deployment. Key to our framework are the number of distributions used in training and the metric used to score the diversity of distribution performance. We have presented several experiments that highlight the challenge of learning many distributions at once. We have additionally presented a technique to group together distributions for

training. Finally, we propose framing the problem of Distribution Shift as analogous to Zero-Shot Task Inference. We believe that rapid advances in Data Augmentation and Generative Modeling will have a large impact on this research direction.

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
