# OpenReview forum: "Multi-Task Distribution Learning"
_ICLR.cc/2022/Conference — ICLR 2022 Submitted_

### Official Review · Reviewer_PK1M · 2021-10-25

**Correctness:** 2
**Technical Novelty And Significance:** 3
**Empirical Novelty And Significance:** 3
**Recommendation:** 3
**Confidence:** 4

**Main Review:**

**Overall thoughts:**

Preparing for or adapting to distribution shift is a an interesting, yet in my opinion understudied, challenge for machine learning models. Real world applications would likely benefit the most from research in this domain, and a work which develops models robust to distribution shift, or able to adapt quickly to such a shift, is likely to be highly celebrated and well received by industry. This paper collapses distributional learning and data augmentations into one perspective and investigates how training models on many "distributions of augmentations" may lead to models which are robust to significant distribution shifts during inference.

**Method**

The goal of this work is to improve a model's robustness to test-time distribution shift by training one model on many different dataset distributions (i.e. data augmentations). After training, this model should perform well on all dataset distribution on which it was trained. The authors judge this is a good indication of robustness to distribution shift and cite Martin Arjovsky to support their framing of the problem (i.e. the requirement to inject prior knowledge about the types of distribution shifts into the training dataset). Later, the authors analyze which distributions should be trained together in one model to increase model generalizability.

It is unsurprising to me that one model can learn multiple dataset augmentations (i.e. "distribution shifts") throughout training and perform well on each of them during test time. This also creates a disconnect from real-world applications of this method in that the distribution shift for a live model is typically unknown (both in terms of what the shift will be and when it will occur). Moreover, simply scaling a model's size may to enable it to learn a robust feature extractor which can handle many different distributions of data.

**Weaknesses and Suggestions for improvement**

Overall, it feels as if this paper was a bit rushed and incomplete. I do not recommend it for acceptance to ICLR, but encourage the authors to invest additional time into crafting the story and improving the empirical analysis for later resubmission. As it is presented, this work feels a bit all over the place, and while the ideas are interesting and promising, they are not developed into a strong or cohesive story.

1. In my opinion, much of the value of distributional shift learning is invested in creating a model which can quickly adapt to changes in the test distribution or is robust to these changes. Table 4 begins to analyze this dimension, but significantly more analysis would be needed to support the authors' method in finding dataset augmentations which would train a model that is especially robust to distribution shift.
2. Similar to point 1, the authors quote Martin Arjovsky to support their framing of the problem (i.e. the requirement to inject prior knowledge about the types of distribution shifts into the training dataset), but this requirement is often unavailable in many real world applications. It  severely limits the scope of this work to the scenarios where a future distribution shift can be anticipated in the present.
3. I would recommend the authors consider refocusing the story of this paper to decide which dataset augmentations to apply to a model to make it robust to corrupted data and test-time distribution shifts. In other words, given ImageNet1k, which dataset augmentations would likely maximize the performance on Stylized ImageNet or ImageNet-Corrupted?

**Strengths**

1. I think the concept of choosing dataset augmentations from a predefined set to create a model which is robust to distributional shifts is very interesting and promising. Given the number of augmentations available to image data, and the pervasiveness of distributional shift for real-world applications, this work addresses an important challenge in computer vision.
2. The introduction and related work contextualize the authors’ method with the most recent work in this domain. It is apparent the authors are familiar with the distributional shift landscape and knowledgeable in this domain.
3. The proposal to develop a "generalization score" disparate from test-time metrics like test loss and test accuracy is also very relevant.

**Nitpicks**

Typos:
 1. Achilles heel => Achilles' heel (should be plural)

A lot of nouns are also capitalized that should not be capitalized. For instance
1. for targeted Generalization => for targeted generalization
2. Data Augmentation => data augmentation
3. Deep Learning => deep learning
4. Distributional Shift => distributional shift
5. Average Distributional Accuracy => average distributional accuracy
6. Distributional Generalization Metrics => distributional generalization metrics

Finally, there are several latex/formatting issues:
1. **?** utilized style randomization to understand intrinsic properties of Convolutional Neural Networks, finding a texture bias => missing reference in latex.
2. Figures and tables have width which exceeds the margins.

**Summary Of The Paper:**

The authors investigate distribution shift between the data seen during training and the data seen during testing by proposing a framework to analyze distribution shift as well as empirical analysis. In particular, they define a training regime for learning from many distributions within one model as well as a lookahead method to determine which distributions should be chosen for a 0-shot distribution shift benchmark.

**Summary Of The Review:**

I think the authors investigate a very interesting/relevant challenge that could be applicable to a wide range of machine learning applications, but the work as it is presented now -- the story, writing, and empirical analysis -- needs significant revision to present a compelling case for the utility, correctness, and applicability of their method.

---

### Official Review · Reviewer_w1qQ · 2021-10-30

**Correctness:** 2
**Technical Novelty And Significance:** 1
**Empirical Novelty And Significance:** Not applicable
**Recommendation:** 1
**Confidence:** 4

**Main Review:**

The paper is more like an undergraduate course project report instead of an academic paper. In this paper, they mainly described how they do, but many important comparisons and studies are totally missing. For example, there is no literature study, no evidence to show the limitation of the literature, and no comparison with state-of-the-art. No motivation is described. Using data augmentation and multi-task learning are widely used techniques in many papers, so they can not be considered a novelty.

**Summary Of The Paper:**

In this work, the authors present a data augmentation method to address the data distribution shift problem in a multi-task learning framework.

**Summary Of The Review:**

Many important comparisons and studies are missing. No technical novelty.

---

> ### Author Response · Authors · 2021-11-12
> **Agree with Reviewer, Thank you for the Review**
>
> Thank you for your time in this review! We agree that the paper needs more work.
>
> We stand by claim that viewing learning with Data Augmentation in a similar light as Multi-Task learning with multiple loss functions is novel.
>
> However, we agree that we have not provided enough evidence or direction for this study.

---

### Official Review · Reviewer_F3K8 · 2021-11-02

**Correctness:** 3
**Technical Novelty And Significance:** 1
**Empirical Novelty And Significance:** 1
**Recommendation:** 1
**Confidence:** 4

**Main Review:**

Pros:

The expression of this paper is okay.


Cons:

The novelty and contribution of this work are marginal.

**Summary Of The Paper:**

This paper presents Multi-Distribution Learning, exploring the effectiveness of Data Augmentation to prepare for Distribution Shift.

**Summary Of The Review:**

The novelty and contribution of this work are marginal.

---

> ### Author Response · Authors · 2021-11-12
> **Mostly Agree with Reviewer, Thank you for the Review**
>
> We agree that this paper needs more work.
> However, we do think that explicitly studying the relationship of train and test distributions and dividing distributions up with the augmentation interface is novel.
> Please refer us to a paper with a similar experimental setup.
>
> Thank you so much for your time and effort in this review!

---

### Decision · Program_Chairs · 2022-01-20

**Decision:**

Reject

**Comment:**

Though some concepts discussed in the submission are interesting, there are many major concerns: there is a lack of literature review, comparison experiments with the state-of-the-art methods are missing, the technical novelty of the proposed method is very limited.
In the rebuttal, the authors agreed with reviewers' comments and did not provide responses to address reviewers' concerns.

Therefore, based on its current form, this submission does not meet the standard of publication at ICLR.